# Determinants of effective interventions for HIV prevention, treatment, and care to address inequitable HIV outcomes among Black Women of African Descent (BWAD) in High-Income Countries: Systematic review protocol

Akalewold Tadesse Gebremeskel[1,2]*, Amoy Jacques[1], Faith Diorgu[1,3], Josephine Etowa[1,2]

1 School of Nursing, Faculty of Health Sciences, University of Ottawa, Ottawa, Ontario, Canada, 2 School of International Development and Global Studies, University of Ottawa, Ottawa, Ontario, Canada, 3 Nigeria Dept of Nursing, Africa Center of Excellence in Public Health and Toxicological Research, University of Port Harcourt, Port Harcourt, Nigeria

* agebr013@uottawa.ca

## Abstract

### Background

In High-Income Countries (HICs) HIV/AIDS continues to disproportionally affect Black Women of African Descent (BWAD) and other racialized groups and is now a major public health concern. Despite the multiple efforts, evidence is limited on the effectiveness of HIV interventions to address the HIV outcomes inequalities among BWAD. This protocol outlines the methodological process of a systematic review that will gather quantitative and qualitative data to examine existing determinants of effective HIV prevention, treatment, and care interventions to address the HIV outcomes disparities and inequities among BWAD in HICs.

### Methods

A systematic review of eligible articles will be conducted using Preferred Reporting Items for Systematic Reviews and Meta-Analysis (PRISMA) guidelines. A comprehensive search of the literature will be made in MEDLINE(R) ALL (Ovid), Embase (Ovid), CINAHL (EBSCO Host), and Global Health (Ovid). Peer-reviewed studies involving the experience of BWAD in HICs; different HIV prevention, treatment, and care interventions both in the community and in a clinical setting; studies that report on the experience of BWAD on HIV intervention/ service including different levels of barriers and facilitators; reports of original research and peer-reviewed articles based on qualitative, quantitative, and mixed study designs published in English from 1980 onwards in HICs will be included. A narrative synthesis, thematic synthesis, and descriptive quantitative analysis of both extracted qualitative and quantitative data will be undertaken.

**Data Availability Statement:** All relevant data are within the manuscript and its Supporting Information files.

**Funding:** This research program is financially funded by the Canadian Institutes of Health Research (CIHR), Grant # FRN 183803. The funder is not responsible for the content of this paper. The funders had no role in study design, data collection and analysis, decision to publish, or preparation of the manuscript and the decision to submit the paper to this journal.

**Competing interests:** The authors have declared that no competing interests exist.

**Abbreviations:** AIDS, Acquired immunodeficiency syndrome; ART, Antiretroviral treatment; BWAD, Black Women of African Decent; COVID-19, Corona Virus Disease 2019; HICs, High Income Countries; HIV, Human Immunodeficiency Virus; JBI, Joanna Briggs Institute; PRISMA-P, Preferred Reporting Items for Systematic Review and Meta-analysis Protocols; PROSPERO, International Prospective Register of Systematic Reviews; SDGs, Sustainable development goals; SDGs, Sustainable Development Goals; UNAIDS, United Nations Programme on HIV/AIDS.

## Conclusion

Substantial changes including tailored interventions are needed to address the inequities in HIV outcomes that disproportionally impact BWAD in HICs. Understanding the determinants of the effectiveness of BWAD-focused HIV interventions is critical to stemming the HIV epidemic and reducing the burden of the disease and poor health outcomes experienced by BWAD in HICs Our study finding will inform the multi level and multisectoral stakeholder including public health, community-based organizations and nongovernmental civil society organization engaged in BWAD HIV and health policy and practice in HICs. Findings from this review will be used to guide effective response to HIV/AIDS using an equity-driven policy and practice framework.

## Trial registration

**PROSPERO registration number:** CRD42023458938.

## Background

Globally in the last four decades HIV, the virus that causes acquired immunodeficiency syndrome (AIDS) is continues to be a major public health concern [1, 2]. According to the Sustainable Development Goal (SDG) 3, the global stakeholders aimed to end the AIDS epidemic by 2030, yet rates of new infections and deaths are not falling rapidly enough to meet that target [3]. Although HIV can affect anyone regardless of race, ethnicity, gender, age, sexual orientation, or where they live, in High-Income Countries (HICs), disparities in HIV risk and prevention and treatment outcomes are becoming an important public health issue. Racialized people, particularly Black Women of African Descent (BWAD)(referred to as Black women and girls) continue to bear a disproportionate burden of HIV/AIDS and related l HIV-related outcomes [1, 2, 4].

Worldwide, significant achievements have been made in HIV prevention, treatment, support, and the reduction of its impact, however, in HICs racialized people, including BWAD, are still experiencing substantial HIV-related outcomes disparities [2, 4]. Since the recognition of HIV/AIDS in 1981, HICs have had their own distinctive epidemic trends, transmission dynamics, and affected subgroups [5, 6]. In those countries, the social and molecular epidemiology of the epidemic has had features relatively different from the utmost impacted regions like sub-Saharan Africa [7].

In HICs, BWADs are disproportionately affected and overrepresented in HIV infection as compared to women of other ethno-racial backgrounds. In the year between 2015 to 2019, the rate of new HIV infections among Black women are eleven times that of White women and four times that of Latina women [7–10]. While BWAD represented 14% of the US population in 2019, they account for 40% of those living with HIV and 60% of new HIV diagnosis among US women in the same year [4, 11, 12]. In the United Kingdom (UK), Ireland, and US, where declines in new HIV diagnoses have been smaller among Black people than among white populations [13, 14]. Similarly, in the UK, BWAD represent 66% of women HIV diagnoses; White women represent only 21% of diagnoses [14]. In Canada, Black people constitute 2.5% of the population but make up 16% of people living with HIV/AIDS in the country. According to the Ontario HIV Epidemiology and Surveillance initiative of women diagnosed with HIV for the first time in 2020, in Ontario (the largest province and where more than 50% of BWAD lives) 44.4% were Black Women, White 31.5% and Indigenous 13% [15–17].

Differences in the presence of disease, health outcomes, or access to care among population groups are called health disparities [18]. Health disparities that are deemed unfair or stemming from some form of injustice in health status or in the distribution of health resources between different population groups, arising from the social conditions in which people are born, grow, live, work and age. Health inequities are systematic differences in the opportunities groups have to achieve optimal health, leading to unfair and avoidable differences in health outcomes [19, 20]. For BWAD population, addressing such health inequities and their causes need robust evidence support and equity-driven policy and practice.

In HICs the causes of these disparities are complex and multifactorial [9, 12, 21, 22]. These disproportionalities observed in high vulnerability rates to HIV is associated with intersecting social determinants, structural and systemic factors (i.e., racism, stigma) and inequitable access to health care that limits achievement of optimal health outcomes [9, 12, 21, 22]. HIV vulnerability among racialized women, and people at the intersection of these identities, is situated in structural contexts of social, economic, and political inequities [19, 20]. Structural factors such as economic insecurity have a complex and indirect association with HIV risk, operating distally to reduce access to HIV testing, prevention, and care [9, 21, 22]. BWAD experience multiple and intersecting barriers compared to non-racialized women when accessing appropriate and responsive health information and care than non-racialized women [9, 22]. Independent of social and structural barriers, multiple factors influence the success of HIV prevention, diagnosis, and treatment cascade including healthcare systems and policy and guidelines [9, 12, 21, 22]. Additionally, the COVID-19 pandemic exacerbated existing health inequalities faced by racial and ethnic minorities including BWAD in HICs (reference needed here)

The United Nations Programme on HIV/AIDS (UNAIDS) announced a new set of ambitious targets in 2020, calling for 95% of all people living with HIV to know their HIV status, 95% of all people with diagnosed HIV infection to receive sustained antiretroviral therapy, and 95% of all people receiving antiretroviral therapy to have viral suppression by 2025 [23]. This replaced the three 90's in 2014 target given several countries reached the target and the emergence of countries reaching coverage levels as high as 95% for testing, treatment and virologic suppression [24].

Although the new HIV infections and AIDS-related deaths have markedly decreased since the peak of the AIDS pandemic, efforts by HICs to address the systematic differences in HIV outcomes among marginalized and racialized people like BWAD have fallen short [2, 25]. Great disparities exist within and between countries to achieve the UNAIDS targets [9, 25]. The existing interventions are critical in preventing new HIV infection and helping people living with virus, however they are insufficient to reach the UNAIDS 95-95-95 targets and ending the health threat of HIV/AIDS in HICs [1, 2] by 2030. The racial and ethnic implications leading to inequities faced by BWAD and racialized women suggest system wide fragmentation regarding HIV prevention programing and policies [25]. In HICs, over the past four decades, unchanged HIV intervention approaches have highlighted an increasingly urgent need for a tailored, innovative, and equitable approach to effectively end the HIV epidemic to address inequalities in HIV-related health outcomes. Beyond the standard approach, substantial changes and efforts are needed to achieve the UNAIDS targets and achieve the goal of HIV/ AIDS elimination while ensuring health equity and equality for disproportionately impacted people including BWAD [1]. Understanding the race and ethnic inequities and determinants of the effectiveness of HIV programs targeting most at-risk and impacted populations like BWAD is critical to stemming the HIV epidemic and HIV outcomes inequality [22, 25]. We argue that reducing HIV-related health outcomes, health disparities and health inequities should be a major focus of the Global AIDS Strategy between 2021 to 2026 [1, 2, 26, 27]. Despite the multiple individual context-based studies focusing on the factors associated with

HIV vulnerability or risk factors, there is limited evidence on the determinants of the effectiveness of HIV intervention to address the HIV inequality outcomes and impact among BWAD in HICs. There is no systematic review focusing on HIV interventions tailored to BWAD and determinants of effective HIV intervention to address the HIV inequality outcomes and impact among BWAD in HICs.

This systematic review will inform practice and policy interventions for effectively addressing HIV and related healthcare as well as improving inequalities in HIV outcomes among BWAD and beyond in HICs. We will identify, evaluate, and summarize the findings of all relevant individual studies on HIV intervention among BWAD in HICs, thereby making the available evidence more accessible for policy and practice for tailored, innovative, and equitable approach to addressing HIV-related Health outcomes inequalities among BWAD and other racialized people.

The plan to end the health threat of HIV/AIDS is determined by evidence-based effective interventions [1, 2, 27, 28]. Due to the unchanged HIV intervention approach in the last four decades, a growing number of evidence have underscored the need for a tailored, innovative, and equitable approach to end the HIV epidemic while addressing HIV-related Health outcomes inequalities [26, 27]. Therefore, it is important to examine determinants of effective HIV prevention, testing, and treatment intervention strategies to address the HIV-related health outcome inequality among BWAD at higher risk of HIV, who still do not have equitable access to prevention, care, and treatment in HICs [4, 10, 22]. We argue that given the complexity of the challenges BWAD and other racial minority community members are experiencing, it is time to expand the current HIV-practices to fully explore the impact of vulnerable groups including BWAD in HICs.

The aim of this protocol is to outline the methodological process of a systematic review that will gather quantitative and qualitative data to examine existing determinants of effectiveness of HIV prevention, treatment, and care interventions to address the HIV outcomes disparities and inequities among Black Women of African descent (BWAD) in HICs.

## The research questions is

What are the determinants of effective interventions for HIV prevention, treatment, and care to improve HIV and related health outcomes among Black Women of African Descent (BWAD) in HICs? More specifically,

1. What are the HIV interventions /programs that are tailored to BWAD in HICs to address the HIV vulnerability and improve health outcomes?

2. What are the barriers and facilitators to providing effective HIV interventions/ initiatives/ program that are tailored to BWAD in HICs?

3. What are the barriers and facilitators to accessing HIV initiatives/program that are tailored to BWAD in HICs?

4. What are key lessons learned to date from efforts to provide or access HIV interventions/ initiatives/program tailored to BWAD HIV and related health needs?

## Methods

### Study design

This systematic review will include quantitative, qualitative and mixed-methods studies addressing the experiences of BWAD.

## Protocol registration and reporting

This systematic review protocol has been registered within the International Prospective Register of Systematic Reviews (PROSPERO): PROSPERO registration number: CRD42023458938.

This systematic review protocol follows the protocol version of the Preferred Reporting Items for Systematic Reviews and Meta-Analyses(PRISMA-P) guidelines for reporting systematic review [29] (see S1 File). This review will be conducted as per the Cochrane Collaboration Handbook of Systematic Reviews [30] and the findings will be reported in accordance with the reporting guidance provided in the PRISMAs statement [31]. Covidence offers tools and a workflow for all data extraction that would be necessary for the final SR reporting. It produces a PRISMA flow diagram and is able to export additional information.

## Inclusion criteria

1. **Population:** We will include studies involving the experience of women and girls self-identified as African or Black Caribbean or Black person who has lived in HICs; Black or having African or Caribbean descent/ origin, and who has lived in the western world for multiple generations such as the descendants of the trans-Atlantic slave trade. Studies' population include both service users and providers in the context of HIV and related interventions/ services in HICs. We defined high-income countries using the 2023 World Bank classification and Organisation for Economic Co-operation and Development (OECD) country membership [32].

2. **Intervention/ Exposure context:** Eligible studies will involve standard specific interventions or combination HIV interventions in both community and in clinical settings. UNAIDS categorized HIV as a mix of different classes of interventions (behavioral, biomedical, and structural) [33]. Behavioral HIV Prevention interventions include education, awareness creation, knowledge creation, pre-exposure prophylaxis (PrEP)/Post-exposure prophylaxis (PEP), biomedical HIV intervention includes diagnosis or HIV testing and counseling (HTC), HIV treatment and care such as Antiretroviral treatment (ART) and prevention of mother-to-child transmission of HIV (PMTCT); Structural HIV intervention include social and economic support aligned with HIV prevention, treatments, and care.

3. **Comparison or control group:** No comparison group for this study

4. **Outcomes of interest:** Primary intervention outcomes: Studies that report on the HIV interventions /programs that are tailored to BWAD in HICs to address the HIV vulnerability and improve health outcomes; Studies that report on barriers and facilitators to providing effective HIV interventions/ initiatives/program that are tailored to BWAD in HICs; and studies that report on barriers and facilitators to accessing HIV initiatives/program that are tailored to BWAD in HICs.
Secondary interventions outcomes: Studies that report on key lessons learned to date from efforts to provide or access HIV interventions/ initiatives/program tailored to BWAD HIV and related health needs.

5. **Study setting and design:** We will include studies based both in the community and in clinical settings on experiences of BWAD in HICs. The eligible study will include non-experimental qualitative, quantitative, and mixed methods study designs and reports of original research and peer-reviewed articles published in English on the experience of BWAD in HICs. The period from 1980 to 2023 thereby accounts for the new wave of research related to HIV and its impact [6].

## Exclusion criteria

Exclusion criteria include if: the study does not include Black women or girls; the study population has non-Black women, Black men, Caucasian participants or contained ethnic minority groups; studies where interventions did not focus HIV intervention; studies that only measured the outcomes of other health disparity, racial disparity, social determinant of health or individual behavior and social inequalities; studies with experimental design, conducted in low- and middle-income countries. Secondary literature (scoping reviews, literature reviews, letters/commentaries, systematic reviews, and meta-analysis), protocols, and case series. Studies do not reporting the full finding like conference abstracts.

## Search methods for identification of studies

**Electronic searches.** A comprehensive search will be conducted using the primary source of literature will be a structured search of major electronic databases. The following databases will be searched: MEDLINE (Ovid), EMBASE and CINAHL for relevant peer-reviewed articles published between 1980 and 2023. The search strategies designed to access published materials comprise of three stages. (i) A limited search of Ovid Medline to identify relevant keywords contained in the title, abstract and subject descriptors. (ii) Terms identified in this way, and the synonyms used by Ovid Medline, EMBASE, and CINAHL will be used in an extensive search of the literature. (iii) Reference lists of the review-eligible full-text articles will be perused to identify more relevant articles. The searches will be designed and conducted by the review team which includes four experienced public health researchers, in collaboration with a Health Sciences librarian, who helped in optimizing the retrieval of relevant citations. We will perform hand-searching of the reference lists of included studies, relevant reviews, or other relevant documents. Content experts and authors who are prolific in the field will be contacted. The search will include a broad range of MeSH terms and keywords related to BWAD-tailored HIV prevention, testing and care, and treatments.

A draft search strategy within Ovid Medline database is provided in S2 File.

## Data collection and analysis

**Selection of studies.** Searches and application of the inclusion/exclusion criteria will be conducted according to the PRISMA flow approach. All the articles(citations) identified by the database searches will be imported into the Zotero citation management software and uploaded in a zip file. The articles retrieved from searches in each database will be uploaded into the Covidence article online management system to be screened by two authors (ATG&JE) within the Covidence database for their relevance and eligibility using the inclusion/exclusion criteria. Covidence has a full blinding for screening and screening conflict resolution. This will include title and abstract screening, followed by full-text screening against the eligibility criteria for studies deemed potentially eligible. We have this section to include steps taken to minimize observer bias and improve inter-rater reliability. E.g. we have clearly indicated that each title, abstract and full article will be screened by two team members and disagreements will be resolved by a third reviewer. Using JBI appraisal processes, we will evaluate any issues of sample selection and representation by primary studies included in our review.

**Data management and extraction.** The searches will be recorded using PRISMA guidelines, including the list of databases searched, recording of the dates (original and updated) searched and the strategies used for each database. The PRISMA (Preferred Reporting Items for Systematic Review and Meta-Analyses) flowchart will be used to document the selection process [34].

The authors will adapt a data collection form based on the needs of the review from a standardized data extraction form by the Cochrane library [29]. A data extraction sheet will be designed to capture information relating to the included articles. Following full-text screening, data will be independently extracted from the retrieved eligible studies by two of the reviewers (ATG and JE). Disagreements will be settled through discussion with a third reviewer (to be assigned). The data extracted will include all details specific to the review question, fulfilling the requirements for a narrative synthesis. This includes the following information from each article: authors and publication year, study setting, and study aim or hypothesis; design and data collection methods, outcome measures; study findings. We will also contact primary study authors for key information when data are ambiguous or missing from the included studies [35–37].

**Certainty of evidence.** We will use the GRADE-CERQual for qualitative studies and GRADE approach for the quantitative studies. The GRADE-CERQual ("Confidence in the Evidence from Reviews of Qualitative research") approach will be applied by two authors independently to appraise and summarize confidence in key findings [38]. GRADE-CERQual approach helps to make judgements on four components: (i) methodological limitations of included studies, (ii) relevance of contributing studies to the research question, (iii) coherence of study findings, and (iv) adequacy of the data supporting the study findings. Judgements related to the four CERQual components will be summarized in a CERQual Qualitative Evidence Profile [38]. We will use the Grading of Recommendations Assessment, Development and Evaluation (GRADE) approach [39] to assess the confidence in the evidence of effectiveness of interventions for HIV prevention, treatment, and care to address inequitable HIV outcomes among BWAD in HICs. We will present our GRADE assessments in a summary of findings table.

## Assessment of risk of bias in included studies

**Appraisal of study quality.** Quality of individual papers will be assessed using an appropriate Critical Appraisal tool. Methodological rigor in this review will be conducted by having two (ATG&JE) independent reviewers critically appraise the methodological validity of the included studies. Differences in the quality assessment will be resolved by discussion among all the authors [35–37]. The discrepancies will be resolved by discussion with the third author. The methodological quality assessments of studies will be performed using Joanna Briggs Institute(JBI) critical appraisal checklists [40]. JBI's critical appraisal tools helps to assess varieties of study design including cross sectional studies, cohort studies, economic evaluations, prevalence studies, qualitative research. The JBI critical appraisal tool includes multiple questions addressing the internal validity and risk of bias of study designs, particularly confounding, selection, and information bias, in addition to the importance of clear reporting [40].

**Ethics and potential amendments.** Ethics approval is not required as the systematic review does not involve the collection of primary data from participants. The collection of data for our review does not involve direct contact with human participants. Instead, we will use published and publicly accessed data. We do not envisage any amendments to the present protocol, but should an amendment be necessary, it will be notified, registered and reported.

**Data synthesis.** Evidence tables of an overall description of the included studies, including data from each paper that provided details of study characteristics like included study setting, country, study design, HIV intervention type, BWAD research participant characteristics, outcomes, and conclusion. A narrative synthesis, thematic synthesis, and descriptive quantitative analysis of both extracted qualitative and quantitative data will be conducted, a method that is ideal for synthesizing evidence from a wide range of research questions and study designs with qualitative, quantitative, and mixed-method approaches, as the emphasis is on an interpretive synthesis of the narrative findings of research [35]. Qualitative studies will be analyzed using

thematic analysis to synthesize and categorize the findings of the included studies into themes (drawing from Braun and Clarke) [41]. Synthesis of data will be described in a narrative synthesis, grouped by study type and participant characteristics and review objective and outcome [35–37]. We will summarize the included studies and findings using UNAIDS categorized HIV interventions (behavioral, biomedical, and structural). Accordingly, determinants of effective HIV intervention strategies to address the HIV outcomes inequalities among BWAD in HICs will aim to inform policy to address the HIV outcomes inequalities and improve the health outcomes of BWAD in HICs.

## Discussion

This protocol outlines the methodological process of a systematic review that will gather qualitative and quantitative data in order to examine the existing experience of BWAD on HIV intervention /service in HICs. Searches and application of the inclusion/exclusion criteria will be conducted according to the PRISMA flow approach. A narrative synthesis, thematic synthesis, and descriptive quantitative analysis of both extracted qualitative and quantitative data will be conducted. First, we will summarize the key findings from the research and link them to the initial research question. Thus, this systematic review will provide an evidence base on multi-level barriers, facilitators and lessons in HIV intervention to inform policy makers and program planners to practice equity driven HIV programs while addressing HIV outcome inequalities among BWAD and other racialized people in HICs. Second, we will place the findings in multi-level context of HIV intervention among BWAD in HICs. By going back to the literature and analyzing how the results fit within previous research, this study will have significant importance in guiding multilevel BWAD context-based public health policy and implementation including using multiprong approach including targeted behavioral, biomedical, and structural interventions. Furthermore, our finding on multilevel determinants of effective HIV interventions will have significant impact in guiding equity-based health policy and intervention to address the HIV vulnerability and improve health of BWAD in HICs. Our study finding will inform the multi level and multisectoral stakeholder including public health, community-based organizations and nongovernmental civil society organization engaged in BWAD HIV and health policy and practice in HICs. Any changes to the protocol will be updated on PROSPERO and final manuscript.

### Strengths and limitations of this study

Strength: First, the best of our knowledge, no systematic review has examined the existing determinants of effectiveness of HIV prevention, treatment, and care interventions to address the HIV outcomes disparities and inequities among BWAD in HICs using quantitative and qualitative studies. Second, the research team has extensive experience in BWAD health and HIV studies and have multiple publications and work on BWAD health and HIV studies in HICs and beyond. Third, the use of a comprehensive search strategy and a range of databases in consultation with health science librarian to develop the protocol.

Limitation, it would be difficult to avoid publication bias because of the exclusion of secondary/grey literature; studies published in languages other than English; and exclusion of studies including population other than BWAD.

### Patient and public involvement

Patients will not be directly involved in the design of the study. As the study is a protocol for a systematic review and no participant recruitment will take place, their involvement in the recruitment and dissemination of findings to participants was not applicable.

### Dissemination of findings

The systematic review and its evidence synthesis will be published in a peer-reviewed journal and presented at different local, national, and international conferences and scientific meetings.

## Supporting information

**S1 File. Preferred Reporting Items for Systematic Review and Meta-analysis Protocols (PRISMA-P).**
(DOCX)

**S2 File. Search terms.**
(DOCX)

## Acknowledgments

The authors are grateful to Collaborative Critical Research for Equity and Transformation in Health (CO-CREATH) Lab, School of Nursing, Faculty of Health Sciences, University of Ottawa for creating enabling situation for Black people health inequity critical research.

## Author Contributions

**Conceptualization:** Akalewold Tadesse Gebremeskel, Josephine Etowa.

**Data curation:** Akalewold Tadesse Gebremeskel.

**Formal analysis:** Akalewold Tadesse Gebremeskel, Josephine Etowa.

**Funding acquisition:** Josephine Etowa.

**Investigation:** Akalewold Tadesse Gebremeskel, Josephine Etowa.

**Methodology:** Akalewold Tadesse Gebremeskel, Josephine Etowa.

**Project administration:** Amoy Jacques, Josephine Etowa.

**Resources:** Akalewold Tadesse Gebremeskel, Amoy Jacques, Josephine Etowa.

**Software:** Akalewold Tadesse Gebremeskel.

**Supervision:** Josephine Etowa.

**Validation:** Akalewold Tadesse Gebremeskel, Josephine Etowa.

**Visualization:** Akalewold Tadesse Gebremeskel, Josephine Etowa.

**Writing – review & editing:** Akalewold Tadesse Gebremeskel, Amoy Jacques, Faith Diorgu, Josephine Etowa.

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
