## [Decision Letter · Decision Letter 0]

12 Dec 2023

PONE-D-23-33845Determinants of effective interventions for HIV prevention, treatment, and care to address inequitable HIV outcomes among Black Women of African Descent (BWAD) in High-Income Countries: Systematic review protocol.PLOS ONE

Dear Dr. Gebremeskel,

Thank you for submitting your manuscript to PLOS ONE. After careful consideration, we feel that it has merit but does not fully meet PLOS ONE’s publication criteria as it currently stands. Therefore, we invite you to submit a revised version of the manuscript that addresses the points raised during the review process.

We look forward to receiving your revised manuscript.

Kind regards,

Yury E Khudyakov, PhD

Academic Editor

PLOS ONE

Journal Requirements:

"The research program is supported by the Canadian Institutes of Health Research (CIHR), Grant # FRN 183803. "

"The authors are grateful to Collaborative Critical Research for Equity and Transformation in

Health (CO-CREATH) Lab, School of Nursing, Faculty of Health Sciences, University of Ottawa

for creating enabling situation for Black people health inequity critical research; and the Canadian 

Institutes of Health Research (CIHR) for providing funding for this study. "

"The research program is supported by the Canadian Institutes of Health Research (CIHR), Grant # FRN 183803. "

**Additional Editor Comments:**

Your manuscript was reviewed by one expert in the field who identified many important problems in your submission and produced copious comments. It is essential that you provide thorough responses to all comments submitted by the reviewer.

Reviewers' comments:

Reviewer's Responses to Questions

**Comments to the Author**

1. Does the manuscript provide a valid rationale for the proposed study, with clearly identified and justified research questions?

Reviewer #1: Yes

2. Is the protocol technically sound and planned in a manner that will lead to a meaningful outcome and allow testing the stated hypotheses?

Reviewer #1: Yes

3. Is the methodology feasible and described in sufficient detail to allow the work to be replicable?

Reviewer #1: Yes

4. Have the authors described where all data underlying the findings will be made available when the study is complete?

Reviewer #1: Yes

5. Is the manuscript presented in an intelligible fashion and written in standard English?

Reviewer #1: No

6. Review Comments to the Author

You may also provide optional suggestions and comments to authors that they might find helpful in planning their study.

Reviewer #1: Dear Authors, 

 

Thank you very much for your hard work and important study protocol for an extremely important subject. Thank you also for the opportunity to read and review your protocol. I believe your study has implications to better inform effectiveness of interventions for HIV prevention, treatment and care to BWAD to address inequitable HIV outcomes in HICs.

Overall, the protocol is good; however, there are several suggestions for sentence structure and grammar for your consideration and I do recommend another peer edit. Please ensure that you stick to one voice and not switch from active to passive voice (i.e., the authors… and then switch to “we will also…” – example in the Data management and extraction section). I have several comments, recommendations and suggestions for your consideration.  Once addressed, I believe the protocol and systematic review will a great contribution.

 

Comments:  

 

Abstract:

- Suggest rewording first sentence (for conciseness) to read: In High-Income Countries (HICs) HIV/AIDS continues to disproportionally affect Black Women of African Descent (BWAD) and other racialized groups.

- Suggest rewording last sentence (for conciseness): This protocol outlines the methodological process of a systematic review that will gather quantitative and qualitative data to examine existing determinants of effective HIV prevention, treatment, and care interventions to address the HIV outcomes disparities and inequities among BWAD in HICs.

Background: 

 

Paragraph 1:

- "Globally in the last four decades HIV, the virus that causes acquired immunodeficiency syndrome (AIDS) is continued to be one of the major public health concerns and health agenda.” Sentence is awkward. Which health agenda? Please clarify the sentence…is it SDGs? Hight income countries? Suggest: Globally in the last four decades HIV, the virus that causes acquired immunodeficiency syndrome (AIDS) continues to be a major public health concern….”

- Racialized people including Black Women of African Descent (BWAD)(referred to as Black women and girls) continue to bear a disproportionate portion of the burden of HIV/AIDS and related l HIV-related outcomes (1,2,4). Sentence is awkward. Suggest "Racialized people, particularly Black Women of African Descent (referred to as Black women and girls) still disproportionately carry the burden of HIV/AIDS and its related outcomes."

Paragraph 2:

- Globally, significant achievements have been registered in HIV prevention, treatment, support, and the reduction of its impact, however, in HICs racialized people including BWAD are still experiencing substantial HIV-related outcomes disparities. Sentence awkward. Suggest: Globally, significant achievements have been made in HIV prevention, treatment, support, and the reduction of its impact, however, in HICs racialized people, including BWAD, are still experiencing substantial HIV-related outcomes disparities.

Paragraph 3

- In the United Kingdom (UK), Ireland, and US, where declines in new HIV diagnoses have been smaller among Black people than among white populations. Suggest taking out “where” to read, “In the United Kingdom (UK), Ireland, and US, declines in new HIV diagnoses have been smaller among Black people than among white populations.”

- In this section, in general, suggest to starting off with general racialized populations, and then focusing on BWAD. Right now it jumps from one to another throughout. Maybe establish the disparities among racialized and Indigenous communities and then focus on BWAD? What do you think? As the population you are focusing on in your systematic review is BWAD.

Paragraph 4

- Please provide a reference for your definition of health disparities.

- Review paragraph as there are some grammatical issues (periods in middle of second sentence (after age).

- The last sentence is quite long. Suggest dividing into two. Stop at “health outcomes (17).” Perhaps the next sentence should read, “They can be addressed by equity driven policy and practice…”. What is meant by policy and practice? Is it decisions, strategies? Please clarify.

Paragraph 5

- Second sentence awkward, suggest “This disproportionately in high vulnerability to HIV is associated with intersecting social determinants, structural and systemic factors (i.e., racism, stigma) and inequitable access to health care that limits achievement of optimal health outcomes.”

- Should be “BWAD experience multiple and intersecting… compared to non-racialized women.” in the 4th sentence.

Paragraph 6

- The following sentence is awkward. “Since the epidemic peaked, although the new HIV infections and AIDS-related deaths have markedly decreased, yet efforts have fallen short for many HICs to address the systematic differences in HIV outcomes among marginalized and racialized people like BWAD.” Not sure what you are meaning. Please clarify, is it that related deaths have decreased since the epidemic peaked (are you referring to the AIDS epidemic?) and yet BWAD continue to bear the burden of HIV/AIDS in HICs? If so, suggest:

“Although the new HIV infections and AIDS-related deaths have markedly decreased since the peak of the AIDS pandemic, efforts by HICs to address the systematic differences in HIV outcomes among marginalized and racialized people like BWAD have fallen short.”?

- Suggest rewriting this sentence: “The existing interventions are critical in preventing new HIV infection and helping people living with virus, however the existing HIV programs are insufficient to reach the UNAIDS 95-95-95 targets and ending the health threat of HIV/AIDS in HICs(1,2) by 2030….. among black and other racialized women” Is this what you are getting at? “The existing interventions are critical in preventing new HIV infection and helping people living with virus, however they are insufficient to reach the UNAIDS 95-95-95 targets and ending the health threat of HIV/AIDS in HICs (1,2) by 2030. The racial and ethnic implications leading to inequities faced by BWAD and racialized women suggest system wide fragmentation regarding HIV prevention programing and policies.”

Paragraph 7

- Awkward wording in the second sentence (‘a growing number of evidence have underscored the need…”). Suggest something like…” "Over the past four decades, unchanged HIV intervention approaches have highlighted an increasingly urgent need for a tailored, innovative, and equitable approach to effectively end the HIV epidemic to address inequalities in HIV-related health outcomes."

Overall comment about the introduction:

You have done a great job providing evidence and rationale behind the disparities faced by racialized women, particularly among BWAD in HICs. The way the introduction is written, you are jumping back and forth between racialized women and then BWAD and sometimes looking at both. Since your research questions is focused on BWAD in particular, may I suggest a funnel system that starts off broadly about HIV, then talk about inequities faced by racialized and Indigenous communities globally, then racialized and Indigenous women globally, then focus on HICs in particular, then focus on BWAD only as that is your focus in this systematic review. What do you think?

Another thought in reading the introduction is the provision of the definitions of health disparities and health inequities in paragraph 4 which almost come out of nowhere and cut the introduction. Suggest somehow, weaving this into the beginning in terms of globally what health dipartites and inequities are and then move into how these impact racialized and Indigenous communities. Whichever way you think flow would be better, as now it seem to cut the flow of thought while reading it.

Systematic Review Question

Currently reads: What are the determinants of effective interventions for HIV prevention, treatment, and care [intervention/comparator] to improve HIV and related health outcomes [outcome] among Black Women of African Descent (BWAD) in HICs [population]? The question is clear in terms of your intervention, population and outcome. The sub-questions provided are well outlined for qual, quant and mixed methods studies. Please note question #3 seems to have “programsthat” instead of “programs that”.

Suggest the following layout for conciseness as there seems to be some overlap in information provided:

- Under study design you describe BWAD and types of studies included. Suggest leaving the population description to the population section in the inclusion criteria aas quite repetitive.

- I would title the first section Protocol registration and reporting and put the section about PROSPERO and PRISMA here right after the major Methods section.

- Next is Inclusion criteria, this is where you describe the population, intervention/exposure, outcomes of interest and types of studies to be included [which is how you have it set up].

- Under Outcomes: In your RQ, you use the word “effective interventions” which insinuates some measurable outcome. How will this be defined? Are there specific primary or secondary interventions outcomes you will be looking at in addition to the sub-questions that focus on the types of interventions, barriers and facilitators and lessons learned. If you use effective, I think you will have to define what you mean by this? What have other protocols defined effective as? Are there examples of outcome variables that showed better health outcomes for BWAD or racialized communities? If so, what are those? Suggest looking at existing protocols/studies that outline effective strategies to guide you.

Table 1: your population states “Adults (16 years or older) who have experienced DSVA; adults who have

perpetrated DSVA”. The question only had adults who experienced DSVA? Should you change it to experience and perpetrate DSVA? It is noted later that you outline (line 170) that experience of DSVA “either as someone who has experienced or perpetrated DSVA”. Since this is not clear from the onset, when reading the question, the phrasing insinuates services/interventions supporting individuals who have experienced DSVA which does not necessarily intuitively include perpetrators. Clarity is needed for the systematic review process and in the question. In the section on intervention/exposure, you list classes of interventions, what were the outcomes of those that you could look at to determine effective ones and how are they deemed effective. I hesitate to leave this section without any examples of possible outcome measures to include in a systematic review. Additional information would provide the reader with a better idea as to what these outcomes are. In this systematic review, will there be measured effects in the quantitative studies? Will these be outlined?

Inclusion/exclusion of studies

- What about letters/commentaries and conference abstracts?

- Indicated only non-experimental; will there be experimental , quasi-experimental, randomized, quasi-randomized trials/studies? You mentioned quantitative, so only non-experimental quantitative. Please clarify.

Search Strategy:

- You have a decent # of databases so am sure you will catch many. Did you consider Gender Watch (ProQuest) [gender and women's studies; higher risk groups] or Web of Science?

- Will there be a grey literature search through google/google scholar? Please clarify rationale for not including grey literature or book chapters/programme reports that maybe relevant.

Certainty of Cumulative Evidence:

- Please clarify, you will use the GRADE-CERQual for qualitative studies and GRADE approach for the quantitative studies?

- Additionally, the citations for both are not the original citations for the GRADE-CERQual and GRADE approach, they are citations of the authors’ own papers when they completed or used this approach. Please use the original citations –

o https://www.cerqual.org/guide-for-decision-makers/

o https://training.cochrane.org/handbook/current/chapter-14

Appraisal of study quality

- Which quality appraisal tool will be used? JBI? CASP? Cochrane ROBINS-I? Please provide rationale as to why this hasn’t been decided.

Discussion:

- Suggest adding a sentence in the discussion that any changes to the protocol will be updated on PROSPERO and final manuscript.

References:

- Note you have the title References twice.

7. PLOS authors have the option to publish the peer review history of their article (what does this mean?). If published, this will include your full peer review and any attached files.

Reviewer #1: No

---

## [Author Response · Author response to Decision Letter 0]

24 Jan 2024

Dear Editor, Dr. Yury E Khudyakov,

 Thank you for giving us the opportunity to submit a revised draft of the manuscript PONE-D-23-33845: Determinants of effective interventions for HIV prevention, treatment, and care to address inequitable HIV outcomes among Black Women of African Descent (BWAD) in High-Income Countries: Systematic review protocol. We appreciate you and the reviewers for your precious time in reviewing our paper and providing valuable comments. The authors have carefully considered the comments and tried our best to address every one of them. We hope the manuscript after careful revisions meet your high standards. The authors welcome further constructive comments if any. 

Below we provide the point-by-point responses, rebuttal letter that responds to each point raised by your self and reviewer(s). We uploaded all the documents including 'Response to Reviewers; 'Revised Manuscript with Track Changes' and an unmarked version of 'Manuscript'.

---

## [Decision Letter · Decision Letter 1]

5 Mar 2024

PONE-D-23-33845R1Determinants of effective interventions for HIV prevention, treatment, and care to address inequitable HIV outcomes among Black Women of African Descent (BWAD) in High-Income Countries: Systematic review protocol.PLOS ONE

Dear Dr. Gebremeskel,,

Thank you for submitting your manuscript to PLOS ONE. After careful consideration, we feel that it has merit but does not fully meet PLOS ONE’s publication criteria as it currently stands. Therefore, we invite you to submit a revised version of the manuscript that addresses the points raised during the review process. Please submit your revised manuscript by Apr 19 2024 11:59PM. If you will need more time than this to complete your revisions, please reply to this message or contact the journal office at plosone@plos.org. Please include the following items when submitting your revised manuscript:A rebuttal letter that responds to each point raised by the academic editor and reviewer(s). You should upload this letter as a separate file labeled 'Response to Reviewers'.A marked-up copy of your manuscript that highlights changes made to the original version. You should upload this as a separate file labeled 'Revised Manuscript with Track Changes'.An unmarked version of your revised paper without tracked changes. You should upload this as a separate file labeled 'Manuscript'.If applicable, we recommend that you deposit your laboratory protocols in protocols.io to enhance the reproducibility of your results. Protocols.io assigns your protocol its own identifier (DOI) so that it can be cited independently in the future. For instructions see: https://journals.plos.org/plosone/s/submission-guidelines#loc-laboratory-protocols. Additionally, PLOS ONE offers an option for publishing peer-reviewed Lab Protocol articles, which describe protocols hosted on protocols.io. Read more information on sharing protocols at https://plos.org/protocols?utm_medium=editorial-email&utm_source=authorletters&utm_campaign=protocols.

We look forward to receiving your revised manuscript.

Kind regards,

Muhammad Shahzad Aslam, Ph.D.,M.Phil., Pharm-D

Academic Editor

PLOS ONE

Journal Requirements:

Reviewers' comments:

Reviewer's Responses to Questions

**Comments to the Author**

1. Does the manuscript provide a valid rationale for the proposed study, with clearly identified and justified research questions?

Reviewer #1: Yes

Reviewer #2: Yes

2. Is the protocol technically sound and planned in a manner that will lead to a meaningful outcome and allow testing the stated hypotheses?

Reviewer #1: Yes

Reviewer #2: Yes

3. Is the methodology feasible and described in sufficient detail to allow the work to be replicable?

Reviewer #1: Yes

Reviewer #2: Yes

4. Have the authors described where all data underlying the findings will be made available when the study is complete?

Reviewer #1: Yes

Reviewer #2: Yes

5. Is the manuscript presented in an intelligible fashion and written in standard English?

Reviewer #1: Yes

Reviewer #2: Yes

6. Review Comments to the Author

You may also provide optional suggestions and comments to authors that they might find helpful in planning their study.

Reviewer #1: Well done on the review of the protocol. The authors have addressed all my comments. One small suggestion for the revised version - under the exclusion criteria, the sentence "Studies don not reporting the full finding article/manuscript report like conference abstracts" reads awkwardly, requires minor editing.

Great and important work and I look forward to reading the final systematic review.

Best wishes.

Reviewer #2: Authors proposed a study of systematic review on the determinants of effective interventions among Black Women of African Descent (BWAD) in high-income countries. Author will conduct the systematic review under the PRISMA-P guidelines. The research topic is important for further understanding sex disparity and race/ethnicity disparity in HIV service. Standard meta-analysis procedures were proposed.

The only major concern is that this proposal is indicating a series of studies, while one same research may contribute information to different sub-analysis regarding different outcome. I.e., it is an umbrella proposal. Intervention strategy and outcome measure of each sub-analysis should be explicitly defined. I suggest authors may consider three points:

1) Describe scope of intervention strategies, and similarly, more specific outcome of interest. A cross-tab like combination table may help audience find specific topic of interest;

2) Propose an example of a specific research target. E.g., effectiveness of PrEP on HIV incidence rate/diagnosis rate among BWAD;

3) Two distinct subgroups among BWAD may experience totally different social economic challenge and support. It will be the best to differentiate new immigrant BWAD from local-born BWAD community.

7. PLOS authors have the option to publish the peer review history of their article (what does this mean?). If published, this will include your full peer review and any attached files.

Reviewer #1: No

Reviewer #2: No

---

## [Author Response · Author response to Decision Letter 1]

12 Mar 2024

Dear Dr. Muhammad Shahzad Aslam, 

Thank you for reviewing and your positive feedback on our manuscript. 

Now, we have addressed all concerns of the reviewers, please see uploaded documents asper your advice.

Thank you for your continued understanding and Support,

Akalewold MSc, PhD©

---

## [Editor Report · Decision Letter 2]

24 Mar 2024

PONE-D-23-33845R2Determinants of effective interventions for HIV prevention, treatment, and care to address inequitable HIV outcomes among Black Women of African Descent (BWAD) in High-Income Countries: Systematic review protocol.PLOS ONE

Dear Dr. Gebremeskel,

Thank you for submitting your manuscript to PLOS ONE. After careful consideration, we feel that it has merit but does not fully meet PLOS ONE’s publication criteria as it currently stands. Therefore, we invite you to submit a revised version of the manuscript that addresses the points raised during the review process.

1-Explain authors prospect interpretation of expected findings, provide context, discuss implications, and suggest future directions.  

2- Highlight how their findings contribute to the current understanding of HIV prevention, treatment, and care among Black Women of African Descent in High-Income Countries (BWAD).

3- Mention that the discussion section typically addresses the implications of the study's findings for theory, practice, and policy. Ask the authors to consider the practical implications of their findings for healthcare providers, policymakers, and researchers working in the field of HIV prevention and care for BWAD.

4- It's still important to address the limitations and strengths of the study that is expected. Planned it properly. 

We look forward to receiving your revised manuscript.

Kind regards,

Muhammad Shahzad Aslam, Ph.D.,M.Phil., Pharm-D

Academic Editor

PLOS ONE

Journal Requirements:

Additional Editor Comments:

1-Explain authors prospect interpretation of expected findings, provide context, discuss implications, and suggest future directions.

2- Highlight how their findings contribute to the current understanding of HIV prevention, treatment, and care among Black Women of African Descent in High-Income Countries (BWAD).

3- Mention that the discussion section typically addresses the implications of the study's findings for theory, practice, and policy. Ask the authors to consider the practical implications of their findings for healthcare providers, policymakers, and researchers working in the field of HIV prevention and care for BWAD.

4- It's still important to address the limitations and strengths of the study that is expected. Planned it properly.

---

## [Author Response · Author response to Decision Letter 2]

6 Apr 2024

Dear Dr. Muhammad Shahzad Aslam thank you for reviewing our systematic review protocol and your positive feedback.

We have addressed your comment by adding more information throughout the manuscript. Please see our response below and the revised manuscript uploaded. 

Akalewold

---

## [Editor Report · Decision Letter 3]

12 Apr 2024

PONE-D-23-33845R3Determinants of effective interventions for HIV prevention, treatment, and care to address inequitable HIV outcomes among Black Women of African Descent (BWAD) in High-Income Countries: Systematic review protocol.PLOS ONE

Dear Dr. Gebremeskel,

Thank you for submitting your manuscript to PLOS ONE. After careful consideration, we feel that it has merit but does not fully meet PLOS ONE’s publication criteria as it currently stands. Therefore, we invite you to submit a revised version of the manuscript that addresses the points raised during the review process.

**1-I have carefully reviewed your manuscript and find it to be a valuable contribution to the field. However, I would like to bring to your attention some potential biases that may need to be addressed. Firstly, it would be beneficial to provide more information about the process of participant selection to ensure the sample is representative of the population under study, thereby mitigating selection bias. Additionally, consider discussing strategies employed to minimize observer bias, such as blinding protocols or inter-rater reliability measures. Furthermore, please ensure that measurement instruments are validated and applied consistently across all study participants to mitigate the risk of measurement bias. Finally, transparency in reporting is crucial. Ensure that all relevant findings, regardless of significance, are included in the manuscript to avoid reporting bias. Addressing these biases will strengthen the validity and reliability of your study findings. Thank you for considering these suggestions, and I look forward to seeing how you incorporate them into your manuscript.**

**2- I commend you on the comprehensive systematic review protocol titled "Determinants of Effective Interventions for HIV Prevention, Treatment, and Care to Address Inequitable HIV Outcomes Among Black Women of African Descent (BWAD) in High-Income Countries." Your protocol lays a strong foundation for investigating crucial aspects of HIV prevention, treatment, and care among BWAD in high-income countries. As I review your protocol, I believe that emphasizing the practical implications of your findings within the discussion section would greatly enhance the relevance and impact of your study. Given the significance of your research topic, it is imperative to clearly articulate how the identified determinants of effective interventions can be translated into actionable strategies to improve HIV outcomes for BWAD. I encourage you to dedicate a section of the discussion to explicitly outline the practical implications of your findings. Consider addressing questions such as: How can the identified determinants inform the development and implementation of targeted interventions for BWAD? What policy changes or healthcare practices could be influenced by your findings to address disparities in HIV outcomes? Providing concrete examples or recommendations for stakeholders and policymakers will strengthen the applicability of your research in real-world settings. By integrating a discussion of practical implications, your study will not only contribute to the academic literature but also provide valuable guidance for addressing healthcare disparities and improving HIV-related outcomes among BWAD in high-income countries.**

**3-I am not satisfied with your current method on Assessment of risk of bias in included studies. Please add Cochrane Collaboration's Tool for Assessing Risk of Bias, Newcastle-Ottawa Scale (NOS) and Jadad Scale.**

**4- Explain Joanna Briggs Institute**

**critical appraisal checklists(40).**

**5.Detail how Covidence will be utilized to assess the risk of bias in the included studies. Describe the criteria and tools that will be used for evaluating study quality and risk of bias, and explain how Covidence will facilitate this process. Describe how data extracted from the included studies will be synthesized and analyzed using Covidence. Outline any planned approaches for meta-analysis, subgroup analysis, or qualitative synthesis, and specify how Covidence will support these analyses. Provide the flow diagram.**

**6- Develop the PICO TABLE.**

**7-It is important to write discussion with previous literature. Please explain why this review is needed.**

We look forward to receiving your revised manuscript.

Kind regards,

Muhammad Shahzad Aslam, Ph.D.,M.Phil., Pharm-D

Academic Editor

PLOS ONE

Journal Requirements:

Additional Editor Comments:

1-I have carefully reviewed your manuscript and find it to be a valuable contribution to the field. However, I would like to bring to your attention some potential biases that may need to be addressed. Firstly, it would be beneficial to provide more information about the process of participant selection to ensure the sample is representative of the population under study, thereby mitigating selection bias. Additionally, consider discussing strategies employed to minimize observer bias, such as blinding protocols or inter-rater reliability measures. Furthermore, please ensure that measurement instruments are validated and applied consistently across all study participants to mitigate the risk of measurement bias. Finally, transparency in reporting is crucial. Ensure that all relevant findings, regardless of significance, are included in the manuscript to avoid reporting bias. Addressing these biases will strengthen the validity and reliability of your study findings. Thank you for considering these suggestions, and I look forward to seeing how you incorporate them into your manuscript.

2- I commend you on the comprehensive systematic review protocol titled "Determinants of Effective Interventions for HIV Prevention, Treatment, and Care to Address Inequitable HIV Outcomes Among Black Women of African Descent (BWAD) in High-Income Countries." Your protocol lays a strong foundation for investigating crucial aspects of HIV prevention, treatment, and care among BWAD in high-income countries. As I review your protocol, I believe that emphasizing the practical implications of your findings within the discussion section would greatly enhance the relevance and impact of your study. Given the significance of your research topic, it is imperative to clearly articulate how the identified determinants of effective interventions can be translated into actionable strategies to improve HIV outcomes for BWAD. I encourage you to dedicate a section of the discussion to explicitly outline the practical implications of your findings. Consider addressing questions such as: How can the identified determinants inform the development and implementation of targeted interventions for BWAD? What policy changes or healthcare practices could be influenced by your findings to address disparities in HIV outcomes? Providing concrete examples or recommendations for stakeholders and policymakers will strengthen the applicability of your research in real-world settings. By integrating a discussion of practical implications, your study will not only contribute to the academic literature but also provide valuable guidance for addressing healthcare disparities and improving HIV-related outcomes among BWAD in high-income countries.

3-I am not satisfied with your current method on Assessment of risk of bias in included studies. Please add Cochrane Collaboration's Tool for Assessing Risk of Bias, Newcastle-Ottawa Scale (NOS) and Jadad Scale.

4- Explain Joanna Briggs Institute

critical appraisal checklists(40).

5.Detail how Covidence will be utilized to assess the risk of bias in the included studies. Describe the criteria and tools that will be used for evaluating study quality and risk of bias, and explain how Covidence will facilitate this process. Describe how data extracted from the included studies will be synthesized and analyzed using Covidence. Outline any planned approaches for meta-analysis, subgroup analysis, or qualitative synthesis, and specify how Covidence will support these analyses. Provide the flow diagram.

6- Develop the PICO TABLE.

---

## [Author Response · Author response to Decision Letter 3]

4 May 2024

Dear Dr. Muhammad Shahzad Aslam thank you for reviewing our systematic review protocol and your positive feedback.

We have addressed your comment by adding more information throughout the manuscript. Please see our response below and the revised manuscript uploaded. 

Akalewold MSc, PhD©

---

## [Editor Report · Decision Letter 4]

9 May 2024

Determinants of effective interventions for HIV prevention, treatment, and care to address inequitable HIV outcomes among Black Women of African Descent (BWAD) in High-Income Countries: Systematic review protocol.

PONE-D-23-33845R4

Dear Dr. Gebremeskel,

We’re pleased to inform you that your manuscript has been judged scientifically suitable for publication and will be formally accepted for publication once it meets all outstanding technical requirements.

Kind regards,

Muhammad Shahzad Aslam, Ph.D.,M.Phil., Pharm-D

Academic Editor

PLOS ONE
---

## [Editor Report · Acceptance letter]

16 May 2024

PONE-D-23-33845R4 

PLOS ONE

Dear Dr. Gebremeskel, 

I'm pleased to inform you that your manuscript has been deemed suitable for publication in PLOS ONE. Congratulations! Your manuscript is now being handed over to our production team.

Kind regards, 

on behalf of

Dr. Muhammad Shahzad Aslam 

Academic Editor

PLOS ONE